# The transforming growth factor beta signaling pathway is critical for the formation of CD4 T follicular helper cells and isotype-switched antibody responses in the lung mucosa

Heather D Marshall[1], John P Ray[1], Brian J Laidlaw[1], Nianzhi Zhang[1], Dipika Gawande[1], Matthew M Staron[1], Joe Craft[1,2], Susan M Kaech[3]*

[1]Department of Immunobiology, Yale University School of Medicine, New Haven, United States; [2]Section of Rheumatology, Department of Internal Medicine, Yale University School of Medicine, New Haven, United States; [3]Department of Immunobiology, Howard Hughes Medical Institute, Yale University School of Medicine, New Haven, United States

**Abstract** T follicular helper cells (Tfh) are crucial for the initiation and maintenance of germinal center (GC) reactions and high affinity, isotype-switched antibody responses. In this study, we demonstrate that direct TGF-β signaling to CD4 T cells is important for the formation of influenza-specific Tfh cells, GC reactions, and development of isotype-switched, flu-specific antibody responses. Early during infection, TGF-β signaling suppressed the expression of the high affinity IL-2 receptor α chain (CD25) on virus-specific CD4 T cells, which tempered IL-2 signaling and STAT5 and mammalian target of rapamycin (mTOR) activation in Tfh precursor CD4 T cells. Inhibition of mTOR allowed for the differentiation of Tfh cells in the absence of TGF-βR signaling, suggesting that TGF-β insulates Tfh progenitor cells from IL-2-delivered mTOR signals, thereby promoting Tfh differentiation during acute viral infection. These findings identify a new pathway critical for the generation of Tfh cells and humoral responses during respiratory viral infections.

*For correspondence: susan.
kaech@yale.edu

**Competing interests:** The authors declare that no competing interests exist.

**Reviewing editor**: Shimon Sakaguchi, Osaka University, Japan

## Introduction

During acute viral infections, CD4 T cells differentiate into primarily T helper 1 (Th1) and T follicular helper (Tfh) effector cells (*Marshall et al., 2011*; *Johnston et al., 2012*; *Hale et al., 2013*). Similar to CD8 T cells, Th1 cells express the transcription factors (TF) T-bet and Blimp1, the effector molecules IFN-γ, TNFα, (and in many cases granzyme B [GrzB] and perforin) and migrate to sites of viral replication to eliminate infected cells. In contrast, Tfh cells primarily remain in secondary lymphoid tissues where they communicate with B cells in germinal centers (GC) to facilitate antibody affinity maturation and isotype switching. Tfh cells express substantially lower levels of T-bet, and instead of Blimp1 they express the TF Bcl6. Some pro-inflammatory cytokines induced during infection, such as IL-12, IFN-γ, IFN-αβ, and IL-2, promote Th1 differentiation; however, the signals required for Tfh differentiation during viral infection have not been as well characterized.

Tfh cells must first encounter their cognate peptide-MHC with proper costimulation from professional antigen presenting cells such as dendritic cells. Following activation, Tfh precursor cells start to express the TF Bcl6 and the chemokine receptor CXCR5, as they downregulate CCR7 and P-selectin glycoprotein ligand 1 (PSGL1) (*Johnston et al., 2009*; *Poholek et al., 2010*; *Choi et al., 2011*; *Pepper et al., 2011*).

**eLife digest** The influenza virus is thought to cause illness in up to 10% of adults and 30% of children each year worldwide. Most of these cases resolve on their own and don't require treatment, but three to five million people are hospitalized and up to half a million people die each year.

Unfortunately, the vaccines currently available to protect against influenza only target particular varieties or "strains" of the virus. The strains that circulate vary from year-to-year so it is necessary to develop new influenza vaccines every year. However, it is difficult to correctly predict which strains will circulate, so a more effective solution would be to develop a new vaccine that can help the body defend itself against many, or ideally any influenza strain.

During a viral infection, a type of immune cell in the host can specialize into two different types of cells to help fight the virus: T helper 1 cells and CD4 T follicular helper cells. T helper 1 cells help to kill host cells that have become infected. CD4 T follicular helper cells promote the production of proteins called antibodies, which identify and neutralize the virus.

Here, Marshall et al. studied how T helper 1 cells and CD4 T follicular helper cells form in mice suffering from a lung infection similar to influenza. It was already known that a protein called transforming growth factor beta (TGF-β) helps the immune response to mount an effective defense against an infection without causing too much harm to the host. Marshall et al. show that TGF-β increases the number of CD4 T follicular helper cells in the mice by suppressing the production of another protein—called IL-2—on the surface of CD4 T cells. Treating mice lacking the ability to detect TGF-β with a drug that blocks a protein controlled by IL-2 also allows more CD4 T follicular helper cells to be produced.

Marshall et al.'s findings reveal that TGF-β is involved in controlling the balance of T helper 1 cells and CD4 T follicular helper cells produced during viral infections of the respiratory tract. Since TGF-β also has other roles in immune responses against viruses, it is now an attractive target for the development of a vaccine that may protect us against all strains of the influenza virus.

These events allow for the migration of activated Tfh precursor cells toward the interfollicular zone and T-B border where they again meet peptide-MHC as well as other costimulatory ligands such as Inducible T cell Costimulator ligand (ICOSL) from B cells (*Breitfeld et al., 2000*; *Schaerli et al., 2000*; *Hardtke et al., 2005*; *Kerfoot et al., 2011*). These interactions result in the further upregulation of Bcl6, migration into the GC, and ability to assist B cells in affinity maturation and proper isotype switching (*Poholek et al., 2010*; *Baumjohann et al., 2011*; *Choi et al., 2011*). In addition to these cell surface ligand-receptor pairings, cytokines play critical roles in the full differentiation of effector Tfh cells during infection. Cytokines utilizing STAT3 signaling pathways including IL-6, IL-21, and IL-27 have been implicated in driving Tfh differentiation, but may have overlapping or compensatory effects depending on the immunizing agent and inflammatory environment (*Ma et al., 2012*; *Ray et al., 2014*). For example, IL-6 appears to act on early anti-viral Tfh precursors (*Choi et al., 2013a*), and while it is not absolutely required for fully differentiated Tfh effector cells during acute lymphocytic choriomeningitis (LCMV) infection (*Poholek et al., 2010*; *Eto et al., 2011*), it does promote the sustained activation of Tfh cells during chronic LCMV infection (*Harker et al., 2011*). Further, IL-27 is required for Tfh differentiation during protein immunization (*Batten et al., 2010*), while IL-21 is sometimes also involved (*Nurieva et al., 2008*; *Vogelzang et al., 2008*; *Eto et al., 2011*; *Karnowski et al., 2012*). In addition to STAT3, STAT4 signaling via IL-12 may also promote early Tfh progenitor cells during infection (*Nakayamada et al., 2011*) and appears to be critical for the differentiation of human Tfh cells (*Schmitt et al., 2009*, *2013*). However, STAT4 signals are absolutely required for the differentiation of Th1 cells, suggesting that additional signals are needed to repress the expression of the Th1 TFs T-bet and Blimp1 in Tfh progenitor cells.

Th1 and Tfh identities can be discerned within the first few days of viral infection indicating that early cytokine signals are involved in the initial stages of the Tfh/Th1 cell fate decision. Increased expression of the high affinity IL-2Rα chain CD25 on early effector CD4 T cells correlates with enhanced expression of the Th1 TFs T-bet and Blimp1 and lower levels of the Tfh TF Bcl6 and this is driven largely by IL-2-STAT5 signaling (*Choi et al., 2011*; *Pepper et al., 2011*; *Choi et al., 2013b*). In contrast, CD25$^{lo}$ early effectors have greater potential to generate Tfh cells (*Ballesteros-Tato et al., 2012*;

*Johnston et al., 2012*; *Nurieva et al., 2012*; *Choi et al., 2013b*). Intriguingly, IL-2 signals are also important for the homeostasis of regulatory T cells (Treg). Therefore, understanding how effector and regulatory CD4 T cells listen to IL-2 will unveil pathways and targets to modulate CD4 T cell responses during infection, autoimmunity, and cancer.

Another important signal at the interface of balancing effector and regulatory CD4 T cells is the cytokine TGF-β. As an immune-suppressive factor, TGF-β promotes the differentiation of peripherally derived regulatory T cells (pTreg) and inhibits the development of autoreactive T cell responses. In contrast, TGF-β can also serve a pro-inflammatory role by inducing the differentiation of effector Th17 cells. T cell-specific ablation of TGF-β signaling, either via TGF-βRII deletion or expression of a dominant negative receptor, has demonstrated that direct TGF-β signals are important for both Treg homeostasis and suppression of effector T cell activation and proliferation (*Li et al., 2006*; *Marie et al., 2006*; *Sanjabi et al., 2009*). The aberrant activation of effector cells in the absence of TGF-β signals cannot be rescued by addition of Treg (*Li et al., 2006*), indicating that direct TGF-β signaling on effector CD4 T cells is required to maintain their homeostasis. Furthermore, TGF-β suppresses T-bet expression (*Gorelik et al., 2002*; *Park et al., 2005*) and the exuberant proliferation of T cells display Th1 attributes (*Ishigame et al., 2013*), demonstrating that TGF-β has the capacity to suppress Th1 differentiation.

In this study, we have identified a new role for TGF-β in balancing the development of Th1 and Tfh cells during acute viral infection. Specifically, we found that CD4 T cell-directed TGF-β was a critical signal for anti-viral Tfh differentiation, GC B cell reactions, and isotype-switched antibody response during influenza infection. TGF-β suppressed the expression of the high affinity IL-2Rα chain CD25, which restricted IL-2 signaling via STAT5 and mTOR in Tfh progenitor cells early during infection in vivo. Finally, we show that blockade of the mTOR signaling pathway can rescue Tfh differentiation of anti-viral CD4 T cells generated in the absence of TGF-β. Thus, we have identified that T cell-directed TGF-β insulates Tfh precursor cells from IL-2 signals and plays a critical role in the generation of effector Tfh cells and high affinity, class-switched antibodies—an essential source of protective immunity to this global health burden.

## Results

### TGF-$\beta$-associated gene expression signature in Tfh cells

To better understand the specification of diverse CD4 T cell subtypes during viral infection, we compared the gene expression profiles of Tfh and Th1 effector CD4 T cell subsets that formed during acute LCMV infection (*Marshall et al., 2011*). This analysis revealed a number of TGF-β-associated genes commonly found in Treg cells, including *Nt5e* (CD73), *Folr4* (folate receptor 4), *Foxp3*, and *Ikzf2* (Helios) (*Hill et al., 2007*), to be more highly expressed in PSLG1$^{lo}$ Ly6C$^{lo}$ T-bet$^{lo}$ CXCR5$^{hi}$ Tfh cells relative to the PSGL1$^{hi}$ Ly6C$^{hi}$ T-bet$^{hi}$ CXCR5$^{lo}$ Th1 cells (*Marshall et al., 2011*; *Hale et al., 2013*) (*Figure 1A*). We first sought to determine if these results indicated that T follicular regulatory (Tfr) cells, a recently described immune-suppressive Tfh population (*Chung et al., 2011*; *Linterman et al., 2011*; *Wollenberg et al., 2011*), formed during acute LCMV infection. To assess this, we infected B6 or TCR transgenic Smarta (Stg) chimeras with acute LCMV Armstrong and monitored Tfh and Treg properties in either GP$_{66-77}$ tetramer$^+$ or Stg CD4 T cells by flow cytometry. Although we detected enhanced *FoxP3* mRNA in the Tfh cells from our microarray analysis, we did not identify any LCMV-specific CD4 T cells that expressed FoxP3 protein or other Treg-associated markers such as GITR, to the level of canonical CD25$^+$ FoxP3$^+$ Tregs (*Figure 1B* and *Figure 1—figure supplement 1*). This suggested that LCMV-specific CD4 T cells do not differentiate into Tfr cells (*Marshall et al., 2011*; *Srivastava et al., 2014*). However, in agreement with the differential mRNA expression, we did find enhanced expression of several of the TGF-β- or Treg-associated proteins including CD73, folate receptor 4, and Helios on Tfh cells relative to the Th1 cells (*Figure 1C*) (*Hill et al., 2007*; *Iyer et al., 2013*). These observations suggested that conventional Tfh cells bear some similarities in their gene expression profiles with Treg cells, despite having little-to-no FoxP3 expression.

### Direct TGF-$\beta$ is a critical signal for Tfh differentiation during acute influenza virus infection

We hypothesized that the expression of these Treg-associated gene products may be an indication of TGF-β signaling in the virus-specific Tfh cells. In order to assess the contribution of direct TGF-β signals

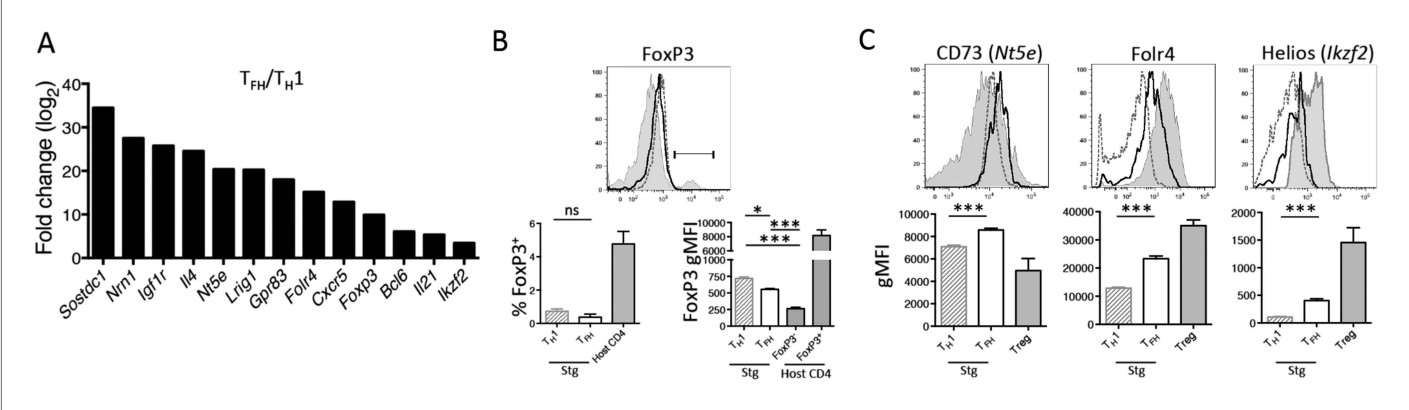

**Figure 1**. TGF-β-associated gene expression signature in Tfh cells. (**A**) Bar graph shows a selected set of genes upregulated in d8 LCMV-specific Stg PSGL1$^{lo}$ Ly6C$^{lo}$ Tfh cells relative to PSGL1$^{hi}$ Ly6C$^{hi}$ Th1 cells isolated and sorted from the spleen as measured using Illumina DNA microarrays (**Marshall et al., 2011**) that have been described to be induced by TGF-β or associated with Treg cells (**Hill et al., 2007**). (**B**) Representative histogram plot (top) shows amount of intracellular FoxP3 in total host splenic CD4 T cells (shaded gray) and LCMV-specific Th1 (hatched line) and Tfh (black line) Stg CD4 T cells from the spleen at day 8 p.i. Region gated identifies FoxP3$^{+}$ nTregs. Bar graphs (bottom) depict the cumulative frequency (left) of FoxP3$^{+}$ CD4 T cells or gMFI averages (right) of the indicated CD4 T cell populations. (**C**) Expression of the indicated Treg-associated proteins in (**A**) was compared between LCMV-specific Th1 (hatched line) and Tfh cells (black line), and FoxP3$^{+}$ Treg cells gated on total host CD4 T cells (shaded gray) from the spleen at day 8 p.i. Histogram plots (top) are representative examples of individual mice and bar graphs (bottom) depict the gMFI averages of each protein in the indicated CD4 T cell populations. Graphs in **B** and **C** are representative of one of five independent experiments (n = 4–5 mice/group/experiment). *p < 0.05, ***p < 0.0005.

The following figure supplement is available for figure 1:

**Figure supplement 1**. LCMV-specific Stg CD4 T cells do not form canonical regulatory T cells nor T follicular regulatory cells.

on the formation of anti-viral CD4 T cell subsets, we crossed TGF-βRII$^{f/f}$ CD4-cre mice to the Stg TCR transgenic mice. Fixing the TCR delays the onset of autoimmunity in the TGF-βRII$^{f/f}$ CD4-cre mice (**Sanjabi and Flavell, 2010**); however, activated CD44$^{hi}$ CD4 T cells do emerge over time (data not shown). Therefore, when making chimeras, we adoptively transferred naïve CD44$^{lo}$ TGF-βRII$^{+/+}$ CD4-cre$^{+}$ Stg cells (herein referred to as WT) or naïve CD44$^{lo}$ TGF-βRII$^{f/f}$ CD4-cre$^{+}$ Stg cells (KO) into congenic C57BL/6 recipients and 1 day later infected the mice with the acute Armstrong strain of LCMV. Intriguingly, we found that direct TGF-β promoted the differentiation of Tfh precursor cells at day 3 post infection (p.i.), such that there were about 1/3 fewer CD25$^{lo}$ CXCR5$^{+}$ Tfh precursor cells in the absence of direct TGF-β signals (WT = 60.25% ± 4, KO = 42% ± 3.3) (**Figure 2—figure supplement 1A**). Additionally, the TGF-βRII KO early effector CD4 T cells expressed more Th1 proteins Ly6C and T-bet and slightly lower Tfh TF Bcl6 (**Figure 2—figure supplement 1B**). However, by day 8 there was no phenotypic difference between TGF-βRII WT and KO CD4 T cells in the spleen (**Figure 2—figure supplement 1C–D**). These data indicated that TGF-β played a role in the early specification of splenic Tfh progenitor cells, but that other signals compensated for TGF-β signaling over the course of a systemic LCMV infection.

Because TGF-β is a dominant regulator of T cells in mucosal tissues, we speculated that it may play a larger role in controlling anti-viral effector T cell responses during infection at mucosal sites, such as the lung. Moreover, respiratory influenza infection induces transcription of TGF-β and the influenza neuraminidase enzyme promotes the cleavage of latent TGF-β complex into its bioactive form in the lung mucosa (**Schultz-Cherry and Hinshaw, 1996**; **Carlson et al., 2010**; **Roberson et al., 2012**). To assess the contribution of TGF-β on the anti-viral CD4 T cell response during a respiratory infection, we infected TGF-βRII WT and KO Stg chimeras i.n. with a recombinant influenza virus expressing the LCMV GP$_{66-77}$ epitope recognized by the Stg TCR (WSN-GP33/66) (**Marsolais et al., 2008**). First, we confirmed that the phenotypic properties of influenza-specific CD4 T cells closely mirrored that of LCMV-specific CD4 T cell populations, and importantly, verified that the influenza-specific Stg cells were neither FoxP3$^{+}$ Treg nor Tfr cells (**Figure 2—figure supplement 2**). Moreover, we found that the proportion and total number of PSGL1$^{lo}$ Ly6C$^{lo}$ and PD-1$^{hi}$ CXCR5$^{hi}$ Tfh cells in the lung-draining mediastinal lymph node (MLN) were markedly reduced in the absence of direct TGF-β signals (**Figure 2A**).

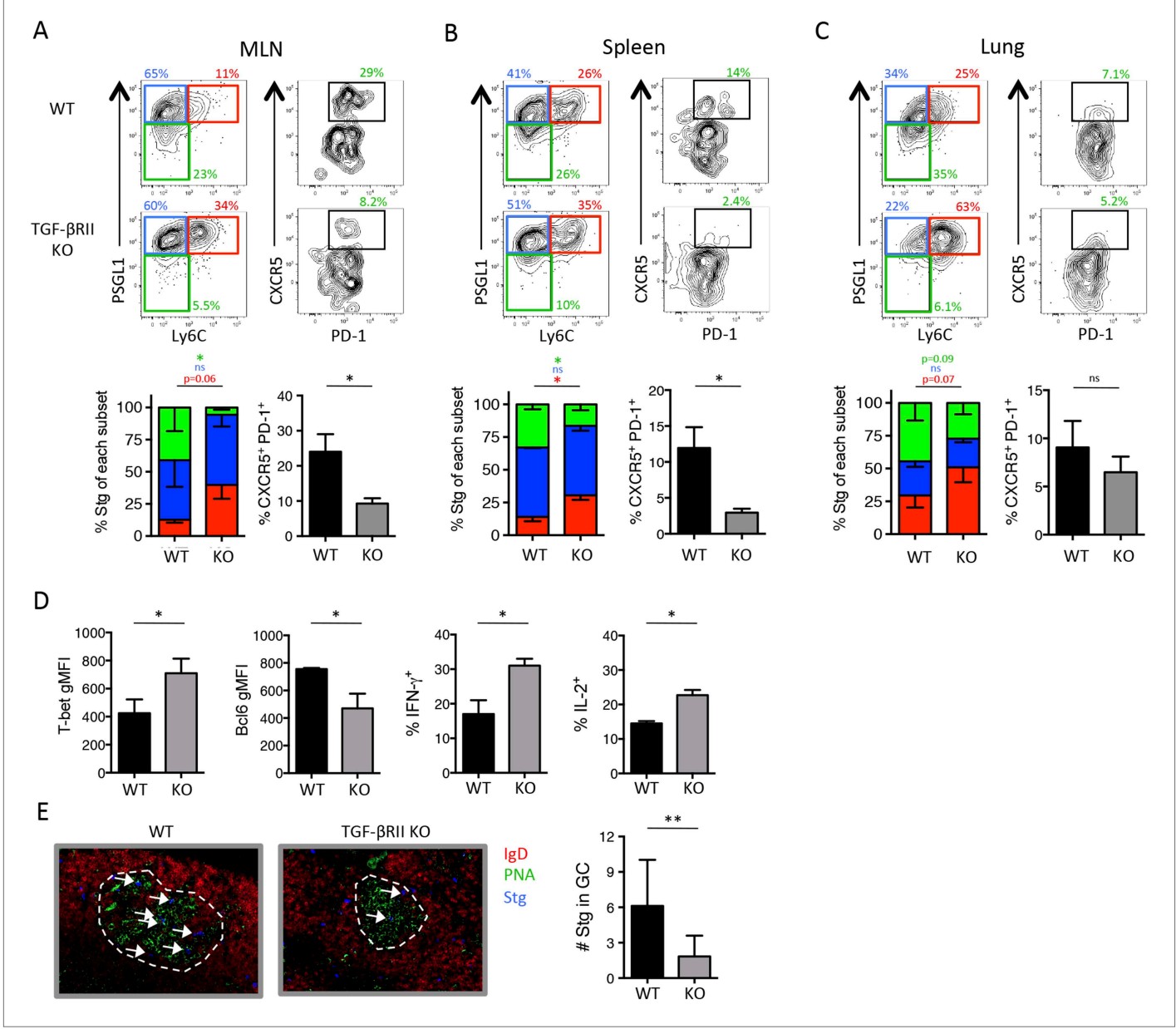

**Figure 2**. Direct TGF-β is required for influenza-specific Tfh differentiation. $2 \times 10^5$ CD44$^{lo}$ TGF-βRII$^{+/+}$ CD4-cre$^+$ (WT) or TGF-βRII$^{f/f}$ CD4-cre$^+$ (KO) Stg cells were adoptively transferred into C57BL/6 congenic recipients infected with WSN-GP33/66 the following day. (**A–D**) On day 8 p.i, Stg cells in the MLN, spleen, and lung were stained with antibodies against the indicated proteins to distinguish Th1 and Tfh cells. Cells were also stimulated with GP$_{66}$-peptide for 6 hr to assess IFN-γ and IL-2 production by intracellular cytokine staining and flow cytometry. (**E**) Stg cells (blue, highlighted by white arrows) located within MLN PNA$^+$ GCs (green) were assessed using immunofluorescent microscopy and their numbers were enumerated using Imaris software. Graphs in **A–D** are representative of one of five independent experiments (n = 4–5 mice/group/experiment). Panel **E** shows representative microscopy images and the cumulative data from two independent experiments with 8 total mice/group were graphed. *p < 0.05, **p < 0.005, and colored asterisks correspond to the color in the stacked graphs.

The following figure supplements are available for figure 2:

**Figure supplement 1**. Direct TGF-β restricts anti-viral T$_H$1 precursor formation but does not impact overall effector CD4 T cell differentiation during LCMV.

**Figure supplement 2**. Expression of PSGL1 and Ly6C distinguish between influenza-specific Th1 and Tfh CD4 T cells.

Furthermore, there was an increase in number of PSGL1$^{hi}$ Ly6C$^{hi}$ Th1 cells in all tissues examined (*Figure 2A–C*). Concomitant with the cell surface phenotypes, we found increased expression of the Th1 TF T-bet and reduced expression of the Tfh TF Bcl6 as well as increased production of IFN-γ and IL-2 in the TGF-βRII KO influenza-specific CD4 T cells compared to their WT counterparts (*Figure 2D*). Finally, and potentially most importantly given their function to help B cells in the GC, we also detected fewer TGF-βRII KO Stg cells localized in PNA$^+$ GC in the MLN (*Figure 2E*), suggesting that TGF-β is important for optimal trafficking of Tfh cells to GCs. Together, these data suggest that direct TGF-β signaling was important for the generation of Tfh cells, while it suppressed Th1 differentiation during respiratory influenza virus infection.

## T cell-directed TGF-$\beta$ signals are required for GC B cell and switched antibody responses

Due to the reduced Tfh differentiation in the absence of direct TGF-β signals, we sought to investigate whether this led to defects in B cell help and formation of antiviral antibodies. Our Stg CD4 T cell adoptive transfer model, as described above, was not suitable to examine GC B cell and antibody responses because the endogenous host-derived CD4 T cells could provide B cell help in these chimeric mice. Therefore, we set up a 'B cell helpless system' using OT-II TCR transgenic mice as recipients because they have a fixed non-influenza-specific CD4 T cell compartment that cannot provide help to GC B cells during the infection. We chose to adoptively transfer polyclonal CD4 T cells in this system to provide a broad repertoire of influenza-specific naïve precursors to provide B cell help. It should also be noted that we switched to the distal Lck-cre deletion strain, which is superior to the CD4-cre strain because of a small number of TGF-βR$^+$ 'escapees' that differentiate into nTreg during neonatal development and prevent autoimmunity in these mice (*Zhang and Bevan, 2012*). Regardless, we set up the experiments similarly to previously described system by adoptively transferring polyclonal naïve CD44$^{lo}$ Thy1.2$^+$ TGF-βRII$^{f/f}$ Lck-cre$^-$ (WT) or TGF-βRII$^{f/f}$ Lck-cre$^+$ (KO) CD4 T cells into congenic Thy1.1$^+$ OT-II recipient mice and 1 day later infected mice with influenza WSN-GP33/66. First, we confirmed that the host-derived OT-II CD4 T cells remained naïve (CD44$^{lo}$) throughout the influenza infection (*Figure 3A–B*). In accord with our findings for the TGF-βRII KO Stg cells, the donor (Thy1.2$^+$) polyclonal activated (CD44$^{hi}$) CD4 T cells lacking TGF-βRII also displayed impaired Tfh cell development. Specifically, in the MLN and spleen, there was a profound reduction in Ly6C$^{lo}$ CXCR5$^+$ cells and a moderate reduction in PSGL1$^{lo}$ Ly6C$^{lo}$ Tfh cells in the TGF-βRII KO cells compared to the WT controls (*Figure 3A–B*). Conversely, there was a concomitant increase in Ly6C$^{hi}$ Th1 cells that lacked TGF-βRII relative to the WT controls.

Importantly, we found that the adoptive transfer of WT CD4 T cells largely restored B cell help in the OT-II recipient mice such that there was an enhanced number Fas$^+$ GL7$^+$ GC B cells and IgM$^{lo}$ B cells in the MLN 14 days p.i. However, the TGF-βRII KO CD4 T cells were unable to rescue the formation of GC B cells in these chimeric mice (*Figure 4A*). Further, WT CD4 T cells partially rescued GC B cell class switching to the IgG1 subtype, while the TGF-βRII KO CD4 T cells did so less efficiently, particularly in the spleen (*Figure 4B*). In addition to the enumeration of GC B cells by flow cytometry, we also observed fewer and smaller GCs from the spleens of mice receiving TGF-βRII KO CD4 T cells by fluorescent microscopy (*Figure 4C*). Finally, T cell-directed TGF-β was also important for influenza-specific IgG and IgA in the airways of infected mice (*Figure 4D*). Taken together, these data demonstrate a previously unappreciated requirement of intrinsic TGF-β signaling in antiviral T cells for Tfh cell function as B cell helpers, and thus, GC reactions and isotype-switched antibody responses during respiratory influenza virus infection.

## Direct TGF-$\beta$ suppresses the formation of Th1 precursors within days of viral infection

Because CD4 T helper subsets begin to diverge within the first few days of viral infection (*Choi et al., 2013b*) and we found fewer Tfh and more Th1 precursor cells in the absence of TGF-β signals during LCMV infection (*Figure 2—figure supplement 1A*), we questioned when TGF-β was required for Tfh differentiation during influenza virus infection. In order to assess this, we generated chimeras with 2 × 10$^6$ CD44$^{lo}$ TGF-βRII WT or KO Stg CD4 T cells, infected the mice 1 day later with WSN-GP33/66 i.n, and assessed the phenotypes of the early effector CD4 T cells in the lung-draining MLN at days 4–5 p.i. Although a difference in Bcl6 expression in the TGF-βRII KO CD4 T cells was not observed at this time point, the TGF-βRII KO CD4 T cells displayed enhanced Th1 attributes including enhanced expression of CD25, Ly6C, T-bet, IFN-γ, and IL-2 (*Figure 5*). These findings suggested that direct TGF-β suppressed Th1 precursor formation within the first few days of influenza virus infection.

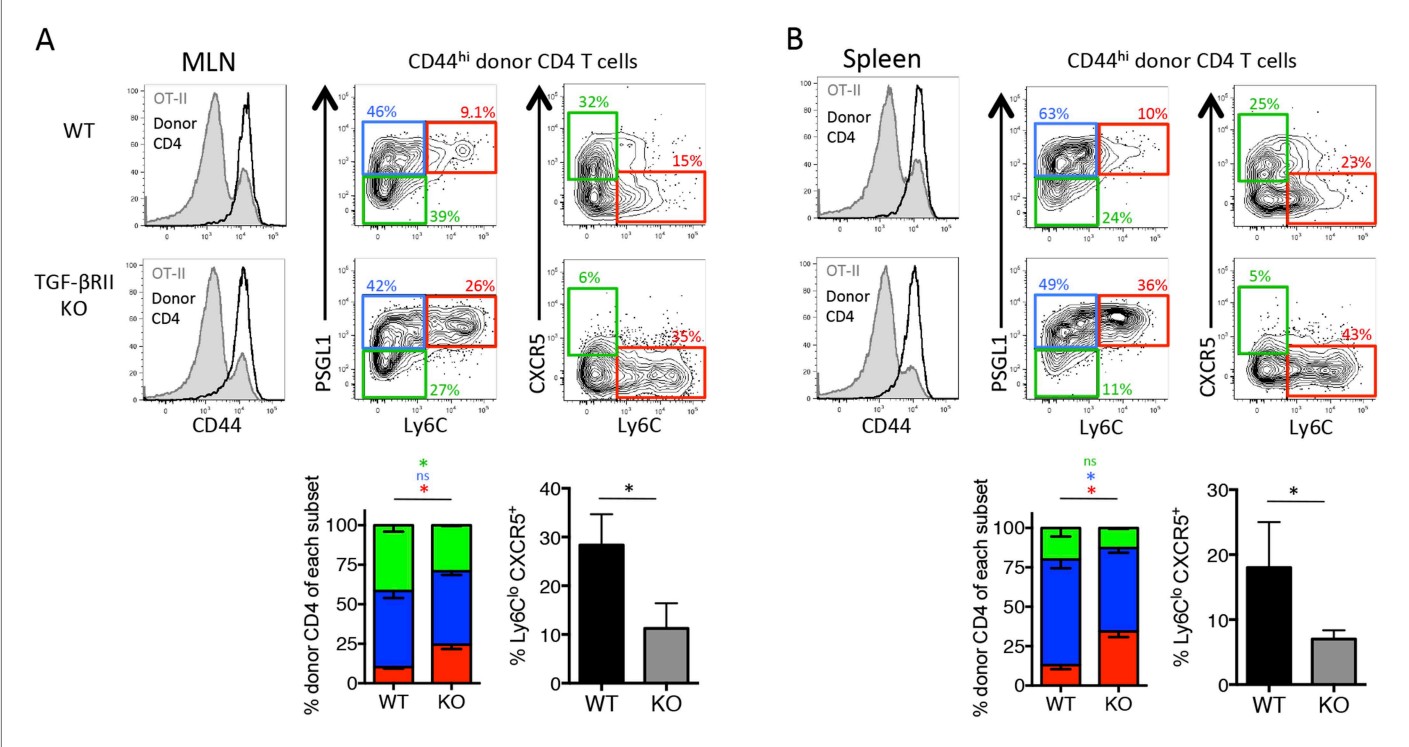

**Figure 3**. Direct TGF-β signaling is required for polyclonal influenza-specific Tfh differentiation. $5 \times 10^6$ CD44$^{lo}$ Thy1.2$^+$ CD4 T cells from TGF-βRII$^{f/f}$ Lck-cre$^-$ (WT) or TGF-βRII$^{f/f}$ Lck-cre$^+$ (KO) mice were adoptively transferred into congenic Thy1.1$^+$ OT-II TCR transgenic mice and infected with WSN-GP33/66 the following day. 14 days p.i., host OT-II (Thy1.1$^+$) and donor (Thy1.2$^+$) CD4 T cells in the MLN (**A**) and spleen (**B**) were assessed for expression of CD44 (to distinguish activated T cells, see histogram plots left) and CXCR5, PSGL1, and Ly6C to distinguish Tfh and Th1 attributes. FACS plots are from representative mice and bar graphs are representative of one of four independent experiments (n = 4–5 mice/group/ experiment). *p < 0.05 and colored asterisks correspond to the color in the stacked graphs.

## Direct TGF-$\beta$ restricts IL-2 responsiveness and insulates early Tfh progenitor cells from mTOR signaling

Due to the enhanced expression of CD25 in the absence of TGF-βRII signaling, we questioned whether TGF-β may modulate the IL-2 responsiveness of early effector CD4 T cells. Addition of recombinant TGF-β (10 ng/ml) to Stg CD4 T cell cultures stimulated with GP$_{66}$ peptide in vitro did not inhibit T cell activation because the upregulation of CD25 and the proliferation rates of the CD4 T cells were comparable between cultures containing or lacking exogenous TGF-β (**Figure 6A**). However, TGF-β profoundly affected the ability of the activated CD4 T cells to sustain CD25 expression and IL-2 responsiveness. That is, at day 2 post activation, the amount of surface CD25 and intracellular phospho-STAT5 (pSTAT5) after IL-2 stimulation was the same whether or not TGF-β was present, but 1 day later, the T cells exposed to TGF-β displayed considerably less CD25 and lower IL-2 responsiveness compared with those that were not (**Figure 6A**). Next, we assessed whether TGF-β modulates CD25 expression and IL-2 signaling in virus-specific CD4 T cells in vivo by comparing CD25, pSTAT5, and pS6 levels in TGF-βRII WT and KO Stg CD4 T cells isolated directly ex vivo from day 3 post LCMV infection, a setting in which Th1 progenitor cells are more frequent to facilitate analysis (**Figure 2—figure supplement 1A**). We observed a considerably enhanced CD25$^+$ pSTAT5$^+$ population from TGF-βRII KO early effector cells relative to the WT cells (**Figure 6B**), suggestive of heightened IL-2 signaling in the TGF-βRII KO CD4 T cells in vivo during infection. Further, we also found enhanced ex vivo CD25$^+$ pS6$^+$ early effector cells (**Figure 6B**), indicating that both STAT5 and AKT/mTOR signaling arms are amplified in the absence of TGF-β signals. Together, these data demonstrate that TGF-β directly suppresses the expression of CD25 and phosphorylation of STAT5 and S6 in early effector CD4 T cells in vivo and thus, likely promotes Tfh cell differentiation by limiting IL-2 signaling in Tfh precursor cells.

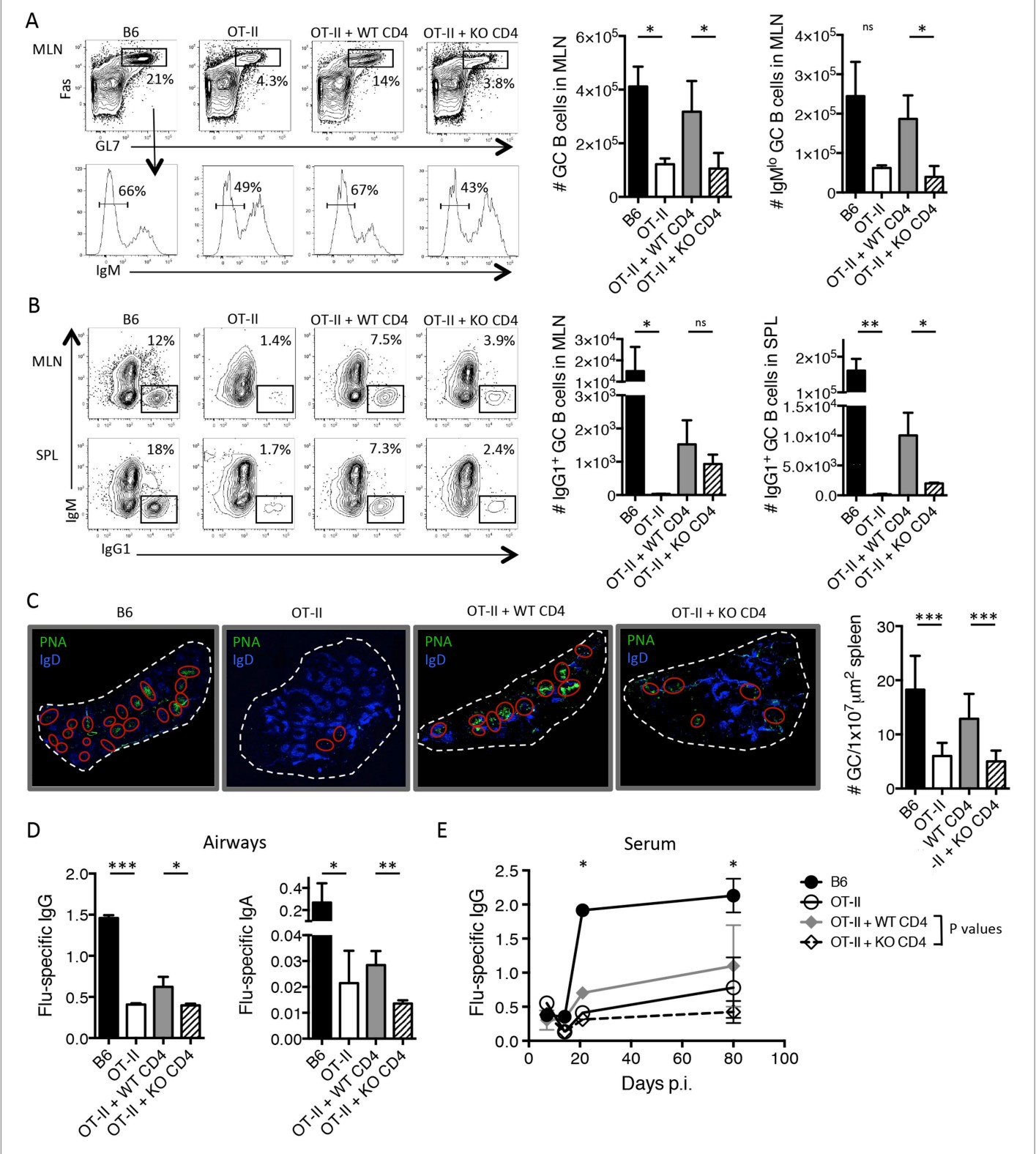

**Figure 4**. T cell-directed TGF-β is required for GC B cell and isotype-switched antibody responses during influenza infection. 5 × 10⁶ CD44^lo Thy1.2⁺ CD4 T cells from TGF-βRII^f/f Lck-cre⁻ (WT) or TGF-βRII^f/f Lck-cre⁺ (KO) mice were adoptively transferred into congenic Thy1.1⁺ OT-II TCR transgenic mice and infected with WSN-GP33/66 the following day. 14 days p.i., GC B cells (Fas⁺, GL7⁺ IgM⁻) (**A**) and IgG1⁺ GC B cells (**B**) in the MLN and spleen (SPL) were

*Figure 4. Continued on next page*

*Figure 4. Continued*

assessed by flow cytometry (left plots) and enumerated in bar graphs (right). (**C**) Splenic PNA⁺ GCs (green, highlighted by red ellipses) were assessed by immunofluorescent microscopy of frozen sections and the numbers of GC/tissue section was calculated using Imaris software. (**D–E**) Influenza-specific IgG and IgA were measured from bronchoaviolar lavage fluid (BAL) by ELISA at day 10 p.i. (**D**) or longitudinally at the indicated time points (**E**). Data in panels **A–B** are representative of four independent experiments (n = 3–5 mice/group/experiment). The bar graphs in panels **C–D** show cumulative data from two independent experiments (n = 3–5 mice/group/experiment), the images in panel **C** are from representative mice of these cohorts. *p < 0.05, **p < 0.005, ***p < 0.0005.

Since we found enhanced pS6 directly ex vivo, we questioned whether we could rescue Tfh differentiation in the absence of TGF-βRII signaling by blocking mTOR activity. To do this, we treated WT and TGF-βRII KO Stg chimeric mice with the mTORC1 inhibitor rapamycin (Rapa) daily throughout influenza infection. It should be noted that IL-2 is not the sole factor that activates mTOR in T cells and that the TCR and a variety of other pro-inflammatory cytokines and costimulatory ligands also utilize this signaling pathway. Interestingly, we found that rapamycin treatment rescued the differentiation of PSGL1ᵸ Ly6Cᵸ Tfh cells and suppressed the aberrant expansion of the PSGL1ʰⁱ Ly6Cʰⁱ Th1 cells that arise in the absence of direct TGF-β signals (*Figure 6C*). Likewise, rapamycin suppressed the over-expression of T-bet and GrzB in the TGF-βRII KO CD4 T cells (*Figure 6D*). Taking into account that mTOR blockade with rapamycin affects many other cell types, these results suggest that TGF-β dampens mTOR activity in CD4 T cells to allow for Tfh cell differentiation. Together with the findings above, these data strongly support that direct TGF-βRII signaling restricts CD25 expression and IL-2 responsiveness in virus-specific CD4 T cells to maximize the development of Tfh cells, GC B cell reactions, and isotype-switched antibody responses during influenza virus infection.

## Discussion

In this study, we demonstrate that T cell-directed TGF-β signals are critical for insulating early effector CD4 T cells from Th1-promoting IL-2 signals, thereby allowing for virus-specific Tfh cell differentiation. Consequently, T cell intrinsic TGF-βRII signaling is also required for GC reactions and isotype-switched antibody production during respiratory influenza virus infection. In support of these findings, TGF-β was also recently shown in human T cells to promote the differentiation of certain Tfh properties in vitro (*Schmitt et al., 2014b*). T cell-directed TGF-β restricted the expression of the high affinity IL-2Rα chain CD25 on early effector CD4 T cells in vitro and in vivo, while, early anti-viral effector cells generated in the absence of TGF-β signals displayed evidence of enhanced IL-2 signaling in vivo. Since IL-2 promotes Blimp1 expression and Th1 differentiation (*Pepper et al., 2011*; *Ballesteros-Tato et al., 2012*;

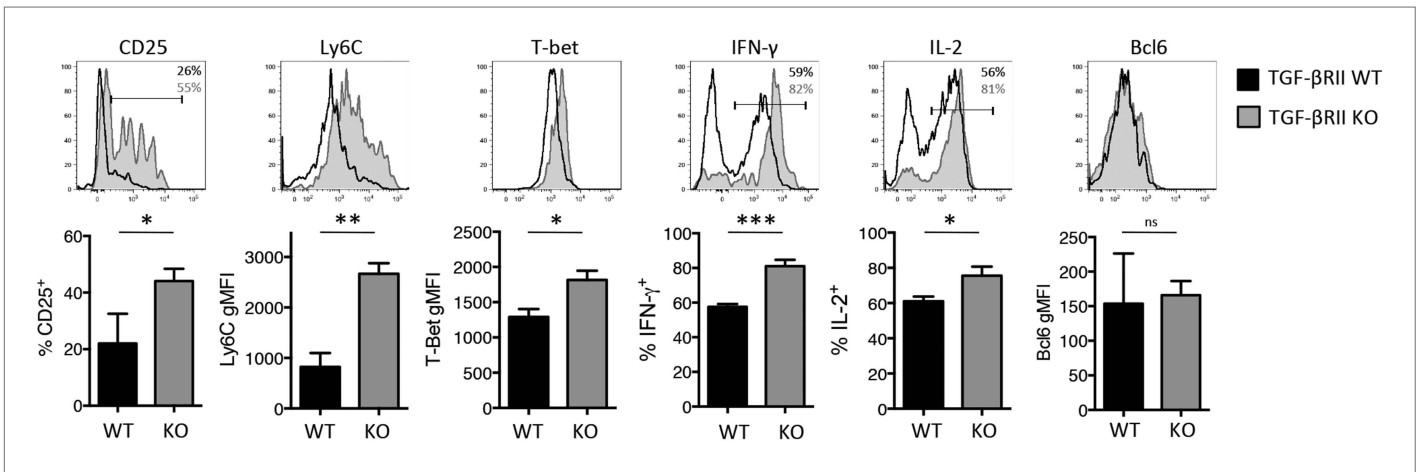

**Figure 5**. Direct TGF-β suppresses early influenza-specific Th1 precursor formation in the lung-draining MLN. 2 × 10⁶ TGF-βRII⁺/⁺ CD4-cre⁺ (WT) or TGF-βRIIᶠ/ᶠ CD4-cre⁺ (KO) Stg cells were adoptively transferred into C57BL/6 congenic recipients that were infected with WSN-GP33/66 the following day. On day 4–5 p.i., CD44ʰⁱ Stg cells in the MLN were assessed for the indicated proteins (CD25, Ly6C, T-bet, IFN-γ, IL-2, Bcl6) by flow cytometry. Cells were stimulated with PMA and ionomycin for 4 hr to assess IFN-γ and IL-2. Bar graphs are representative of three independent experiments (n = 3–4 mice/group/experiment). *p < 0.05, **p < 0.005, ***p < 0.0005.

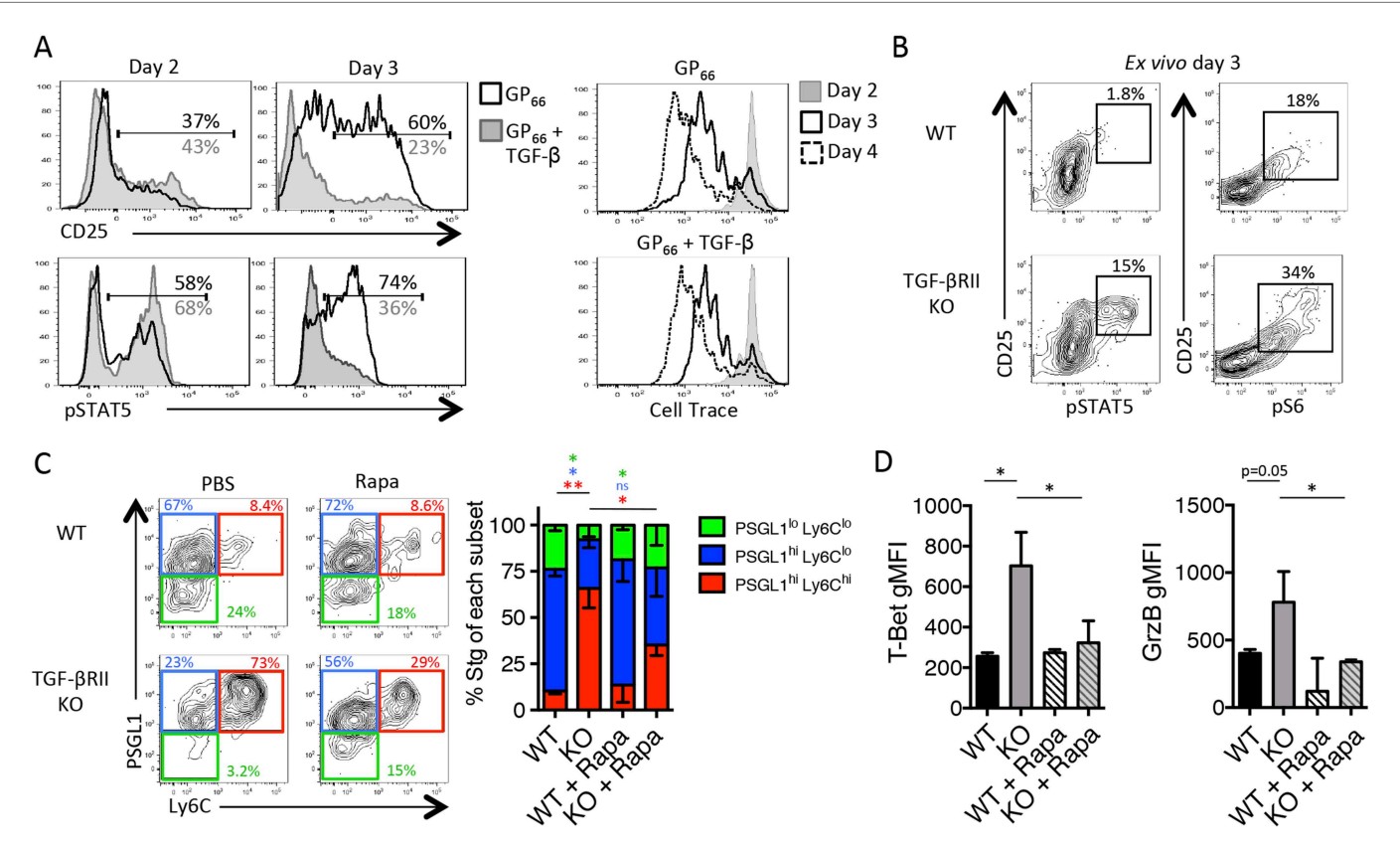

**Figure 6**. TGF-β restricts IL-2 responsiveness and insulates early Tfh progenitor cells from mTOR signaling. (**A**) Stg CD4 T cells were labelled with Cell Trace dye and cultured in vitro with 0.1 μM GP66 peptide ± 10 ng/ml TGF-β and stained for surface expression of CD25 (top) or restimulated with IL-2 to assess pSTAT5 (bottom). Data are representative of four independent experiments. (**B**) Stg chimeric mice (1 × 10⁶ CD44lo TGF-βRII+/+ CD4-cre+ (WT) or TGF-βRIIf/f CD4-cre+ (KO)) were infected with LCMV Armstrong and 3 days p.i., spleens were fixed immediately in 2% PFA and stained with antibodies to measure surface expression of CD25, Ly6C, and intracellular phosphorylation of STAT5 (pSTAT5) and pS6 directly ex vivo. Data are representative of three independent experiments including 3–5 total mice/group. (**C–D**) 2 × 10⁵ CD44lo TGF-βRII+/+ CD4-cre+ (WT) or TGF-βRIIf/f CD4-cre+ (KO) Stg cells were adoptively transferred into congenic C57BL/6 recipients infected with WSN-GP33/66 the following day. Mice were treated with PBS or 75 mg/kg rapamycin i.p. daily. On day 8 p.i, Stg cells in the MLN were assessed for expression of PSGL1 and Ly6C (**C**) or T-bet or GzmB (**D**). Data are representative of three independent experiments encompassing a total of 9–15 mice/group. *p < 0.05, **p < 0.005 and colored asterisks correspond to the color in the stacked graphs.

*Johnston et al., 2012*; *Nurieva et al., 2012*; *Oestreich et al., 2012*), these data suggest that TGF-β works to insulate early effector CD4 T cells from IL-2 signals and may play a central role in limiting Blimp1 expression in Tfh progenitor cells. Interestingly, TGF-β signaling in B cells also promotes IgA class switching and mucosal immunity (*Cazac and Roes, 2000*; *Borsutzky et al., 2004*; *Seo et al., 2013*). Since TGF-β is important for the differentiation of both mucosal Tfh and B cells, which also must interact for proper antibody responses, this provides a unique example of a coordinated signaling pathway to maximize humoral immunity to respiratory influenza virus infection.

Cytokines utilizing the STAT3 signaling pathway including IL-6, IL-21, and IL-27 have been implicated in driving Tfh differentiation (*Nurieva et al., 2008*; *Batten et al., 2010*; *Eto et al., 2011*; *Ma et al., 2012*; *Choi et al., 2013a*; *Harker et al., 2013*; *Ray et al., 2014*), but due to their partially compensatory pathways, separating the requirements for individual signals throughout infection has proved challenging. Furthermore, it is unclear whether other signals induced locally or globally during viral infections may contribute to Tfh cell differentiation. Herein, we identify TGF-β as an additional signal that may restrict IL-2-induced Blimp1 and directly suppress T-bet expression to allow for Tfh cell differentiation. STAT3 signaling in the presence of TGF-β is also a well-described inducer of Th17 effector cells. In fact, a subset of human Tfh cells share multiple properties with that of Th17 cells including co-expression of Bcl6 and Rorγt and the ability to provide B cell help (*Schmitt et al., 2014a, 2014b*).

However, we did not detect Rorγt expression in murine anti-viral effector CD4 T cells (data not shown). Moreover, addition of TGF-β to murine T cell cultures in the presence of IL-6 and IL-21 induced IL-17-producing T cells as expected, but actually suppressed Tfh properties including ICOS and IL-21 expression (*Schmitt et al., 2014b*). These data suggest that although TGF-β and STAT3 signals are required for murine Tfh differentiation during viral infection in vivo, they are not sufficient to induce this cell fate in vitro and that the specification of Tfh cells by these factors is under tight regulation in vivo. Since STAT4 signaling via IL-12 can promote early Tfh properties including Bcl6 expression (*Nakayamada et al., 2011*), it is possible that IL-12-STAT4 signaling is also required to induce a low level of T-bet in murine Tfh precursor cells to prevent Th17 development, but this will have to be formally evaluated.

In the absence of TGF-β, blocking mTOR with rapamycin during the first week of infection was sufficient to restore Tfh cell differentiation and suppress T-bet and GrzB expression. However, rapamycin treatment had no overt affect on the differentiation of WT Stg cells during this phase of influenza infection. Interestingly, rapamycin was recently reported to promote heterosubtypic immunity to influenza by reducing GC reactions and switched antibody responses, resulting in enhanced levels of protective influenza-specific IgM antibodies (*Keating et al., 2013*). This study showed that the protective effects of rapamycin depended on both CD4 T and B cells, and that B cell-intrinsic mTORC1 activity was responsible for enhancing *Aicda* expression and class switching. Although the generation of Tfh cells was not examined in this prior study, our new findings would suggest that suppression of mTOR in the virus-specific CD4 T cells would likely suppress Th1 in favor of Tfh cell differentiation, possibly contributing to the protective effects of rapamycin on heterosubtypic immunity to influenza.

Intriguingly, Treg cells appear to play a central role in anti-viral effector CD4 T cell fate decisions during viral infections due to their production of TGF-β and ability to consume IL-2. In support of this hypothesis, Treg cells were recently shown to be required for Tfh differentiation and GC reactions during influenza infection (*Leon et al., 2014*). Although this group did not find a role for TGF-β by treating mice with an anti-TGF-β blocking antibody, we have demonstrated a requirement for direct TGF-β using genetic ablation of the signaling receptor. Thus, Treg cells appear to play a central role in effector CD4 T cell fates during viral infection, likely by controlling the local bioavailability of both IL-2 and TGF-β. Another potential consideration is the differentiation of influenza-specific pTreg cells, which can temper pulmonary inflammation upon secondary or heterologous infections (*Brincks et al., 2013*; *Kraft et al., 2013*). Future studies will determine whether influenza-specific pTreg, naturally occurring nTreg cells, or other cell types altogether, are a physiologically relevant source of TGF-β to promote Tfh cell differentiation. Further, it is unclear at this time how TGF-β may be interpreted differently by anti-viral Tfh cells and Treg cells, which can suppress CD25 in the former context, but not in the latter. It is likely that IL-2 itself is playing an important role in this setting by initiating a dominant positive feedback loop involving IL-2, STAT5, Blimp1, and FoxP3 to maintain the full Treg program in pTreg cells (*de la Rosa et al., 2004*; *Fontenot et al., 2005*) and that TGF-β may only be required for the initial induction of FoxP3, but this remains to be addressed.

Infectious pathogens such as influenza virus are a global health burden, and despite annual modifications to seasonal flu vaccines that induce protective antibody responses, we have stumbled in our quest for a universal vaccine. Ideally, a universal flu vaccine would elicit broadly protective circulating and lung-resident memory T cells as well as circulating IgM, IgG, and mucosal IgA antibodies that target conserved viral structures to maximize the potential for immunological cross-reactivity to drift variants and different viral subtypes. Since TGF-β plays a critical role in mucosal tissues in the formation of tissue-resident memory CD8 T cells (*Mackay et al., 2013*; *Zhang and Bevan, 2013*), IgA-producing B cells (*Cazac and Roes, 2000*; *Borsutzky et al., 2004*), and as revealed here Tfh cells, it is an attractive target to consider in the development of a broadly protective universal influenza vaccine.

## Materials and methods

### Mice and infections

C57BL/6Ncr mice were purchased from the National Cancer Institute (Frederick, MD). TGF-βRII$^{f/f}$ CD4-cre mice were obtained from R Flavell and crossed to Stg TCR transgenic mice. TGF-βRII$^{f/f}$ Lck-cre mice were a gift from M Bevan. Mice were infected with $2 \times 10^5$ pfu LCMV i.p. or ~50 pfu recombinant influenza WSN-GP33/66 provided by Dr Michael Oldstone (*Marsolais et al., 2008*) i.n. after anesthetizing with ketamine hydrochloride and xylazine. Rapamycin-treated mice were administered ~75 mg/kg rapamycin i.p. daily. All animal experiments were done with approved Institutional Animal Care and Use Committee protocols.

## Cell isolation and adoptive transfers

At various time points post infection, SPL, MLN, bronchoaviolar lavage (BAL), and lungs were dissected. SPL and LN were processed as previously described (*Marshall et al., 2011*). BAL samples were collected by flushing the lungs twice with PBS and collecting both the supernatant for antibody ELISA and cell fraction for flow cytometry. Lymphocyte isolation from the lung tissue was achieved with the Miltenyi Biotec MACSDissociator using their published protocols. Direct ex vivo phospho-staining was performed by homogenizing the spleen in 2% paraformaldehyde immediately after isolation and permeabilizing the splenocytes in ice-cold methanol. To make Stg chimeras, splenocytes were isolated from WT or TGF-βRII KO Stg mice. $CD44^{hi}$ cells were depleted by staining the cells with an anti-CD44 biotin antibody, followed by labeling with the EasySep Biotin selection reagent (Stem Cell Technologies, Vancouver, Canada). The $CD44^{hi}$ bead-bound fraction was removed by placement in a magnet. Purity of depletion was assessed by streptavidin staining, and cells were used only if the $CD44^{hi}$ fraction was <5%. $1 \times 10^4$ Stg cells were adoptively transferred via retro-orbital injection for day 8 LCMV infection, $1 \times 10^6$ for day 3 LCMV, $2 \times 10^5$ for day 8 influenza infection, or $2 \times 10^6$ for days 4–5 influenza infection. Polyclonal chimeras were made in the same fashion and $5 \times 10^6$ $CD44^{lo}$ CD4 T cells were adoptively transferred into OT-II recipient mice.

## Surface and intracellular staining

Lymphocyte isolation and surface and intracellular staining were performed as described previously (*Marshall et al., 2011*). For in vitro stimulation, lymphocytes were stimulated with $GP_{66-77}$ peptide (1 µg/ml) for 6 hr with Brefeldin A and 10 ng/ml IL-2 or 1 µg/ml PMA and ionomycin. $GP_{66-77}$ MHC class II tetramer (NIH tetramer core facility, Emory University, Atlanta, GA) staining was performed at 37°C for 1.5 hr. CXCR5 staining was achieved by incubating the cells at 25°C for 1 hr. TF staining was performed after permeabilization with the FoxP3 fixation and permeabilization kit (eBioscience). Phospho-flow was conducted by stimulating lymphocytes with soluble rIL-2 for 25 min at 37°C, fixing cells in 2% paraformaldehyde, and permeabilizing with ice-cold methanol. Antibodies were purchased from Biolegend (San Diego, CA), BD Pharmingen (San Diego, CA), eBioscience (San Diego, CA), or Cell Signaling Technology (Danvers, MA). Flow cytometry was acquired with a BD LSRII with Diva software and analyzed with Flow Jo software (Treestar, San Carlos, CA).

## In vitro CD4 T cell cultures

Total splenocytes from Stg mice were cultured in RPMI supplemented with 10% FCS, 1% L-glutamine, 1% pen-strep, and 50 µM β-mercaptoethanol plus 0.1 µg/ml $GP_{66-77}$, 10 ng/ml recombinant human TGF-β (Peprotech) was added to specified wells.

## Immunofluorescence microscopy

Tissues were fixed in 4% paraformaldehyde overnight, followed by sinking in sequentially greater sucrose solutions (10%, 20%, and 30%). Fixed tissues were embedded in OCT compound (Sakura), and tissue blocks were frozen in 2-methylbutane (Sigma–Aldrich) chilled by dry ice. Eight micrometer sections were cut with a cryostat, air-dried, and fixed with cold acetone. Sections were stained with 1–5 µg/ml antibodies against Ly5.1, CD4, PNA, IgD, and ProLong Gold antifade reagents (Invitrogen) was added after washing. Images were captured on a Zeiss LSM 510 Meta confocal microscope, mounted on an Axiovert 100 M with automated XYZ control equipped with an argon laser with emissions at 458, 488, and 514 nm and two HeNe lasers with emission wavelengths at 543 and 633 nm. Image analysis was performed using Imaris suite (Bitplane, South Windsor, CT).

## Antibody ELISA

96-well Polysorp microtiter plates (Nunc) were coated overnight with UV-inactivated WSN-GP33/66 in carbonate buffer. AP-conjugated goat anti-mouse IgG and IgA secondary antibodies were used for detection (Southern Biotech). ODs were converted to units based on standard curves with sera from C57BL/6 mice infected with influenza (Softmax Pro 3.1 software; Molecular Devices).

## Statistical analysis

Where indicated, p values were determined by two-tailed unpaired Student's *t* test. p values <0.05 were considered significant and denoted as *$p < 0.05$, **$p < 0.005$, and ***$p < 0.0005$. All error bars represent standard deviation.

## Acknowledgements

The authors would like to thank all the members of the Kaech and Craft labs for their comments and advice on this study, Drs Richard Flavell and Michael Bevan for mice, and Dr Michael Oldstone for influenza virus. This work was supported by NIH grants F32-AI094791 and T32-AI007019-35 (HDM), R01-AI074699 (SMK), AR40072-24, AR053495-08, and the Alliance for Lupus Research (JEC).

## Additional information

### Funding

| Funder | Grant reference number | Author |
| --- | --- | --- |
| National Institutes of Health | F32-AI094791 | Heather D Marshall |
| National Institutes of Health | T32-AI007019-35 | Heather D Marshall |
| National Institutes of Health | RO1-AI074699 | Susan M Kaech |
| National Institutes of Health | AR40072-24 | Joe Craft |
| National Institutes of Health | AR053495-08 | Joe Craft |
| Alliance for Lupus Research | | Joe Craft |

The funders had no role in study design, data collection and interpretation, or the decision to submit the work for publication.

### Author contributions

HDM, Conception and design, Acquisition of data, Analysis and interpretation of data, Drafting or revising the article; JPR, Conception and design, Acquisition of data; BJL, NZ, DG, Acquisition of data; MMS, JC, Conception and design; SMK, Conception and design, Analysis and interpretation of data, Drafting or revising the article

### Ethics

Animal experimentation: This study was performed in strict accordance with the recommendations in the Guide for the Care and Use of Laboratory Animals of the National Institutes of Health. All of the animals were handled according to approved institutional animal care and use committee (IACUC) protocols (2013-10806) of Yale University School of Medicine. The protocol was approved by the Committee on the Ethics of Animal Experiments of the Yale Animal Resource Center (YARC) at Yale School of Medicine.

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
