## [Decision Letter]

Thank you for sending your work entitled “TGF-β signaling is required for the formation of CD4 T follicular helper cells and isotype-switched antibody responses” for consideration at *eLife*. Your article has been favorably evaluated by Tadatsugu Taniguchi (Senior editor), a Reviewing editor, and 2 reviewers.

The Reviewing editor and the other reviewers discussed their comments before we reached this decision, and the Reviewing editor has assembled the following comments to help you prepare a revised submission.

How Tfh cells, which are a critical helper population in the induction and maintenance of germinal center responses, are controlled in their development is only poorly understood at this time. This makes the results in this report timely and of potentially significant interest to the field. In particular, the role of TGF-β and IL-2/CD25 in the formation of Tfh cells is interesting. However, the data demonstrating the role of TGF-β in CD25 downregulation may need to be strengthened, particularly given the well described role of TGF-β in induction of CD25 positive iTregs. As described below, specific issues including some caveats associated with use of the CD4-cre TGF-βRII^f/f^ mice need to be addressed in revising the manuscript.

1) Tgf-β is also well established as a factor important for the in vitro generation of Foxp3^+^CD25^+^ iTregs and in vivo pTregs from CD4+CD25- cells. In these conditions, Tgf-β is perhaps not directly responsible for inducing CD25 expression, but also does not suppress CD25 expression as iTregs and pTregs both express and are dependent on CD25 expression and IL-2 for their survival. However, in Figure 6, it is clear that CD25 expression is being suppressed by Tgf-β. This discrepancy between the role of Tgf-β in the induction of CD4+CD25+Foxp3^+^ T-cells and the blocking of CD25 expression seen here needs to be addressed. Is it due to the Stg TCR, which, as the authors demonstrate, is not amenable to induction of Foxp3 positive cells?

2) In Figure 6, does this loss of CD25 expression still occur if purified naive T-cells are used alone with polyclonal stimuli such as anti-CD3 and anti-CD28? It is obscure to what extent the effect of the TFGb is direct to the CD4 cells or via other cells in the culture, which includes total splenocytes. This is the most direct demonstration that the effect is due to downregulation of CD25 in particular and not more general changes to activation. Please also include the proliferation data, currently not shown, in Figure 6, that Tgf-β is not affecting activation is a key point.

3) The authors describe the immunological events following viral infection in detail but do not describe any effect on the natural history of viral infection. Are OT-II mice transferred with TGFRII KO or WT CD4 more or less able to resolve infection? Given the recent finding that rapamycin decreases germinal center formation and IgG production but enhances protection from influenza, it is not necessarily clear if the observed effects would be beneficial or detrimental to protection from influenza. The authors suggest that part of the contribution of rapamycin to protection seen in the previous study may have been in enhancing Tfh function but it is hard to see how this matches with the reduction in IgG and GC formation. This will make it possible to make more secure statements regarding the in vivo importance of the work.

4) The authors interpret their data as evidence that Tgf-β signaling is “required” for Tfh. Can this be justified given that most of their data assesses the relative representation of Tfh versus Th1? Is the major phenomenon being observed that Tgf-β inhibits the Th1 response?

5) How do the authors explain the observation that “Tfh precursor cells” are impaired at d3 but normal by d8 (Figure 2—figure supplement 1)? Why do other signals compensate for Tgf-β in the setting of LCMV infection but not influenza?

6) The B cell analysis (Figure 4) is very clear, perhaps more striking than the Tfh data (particularly where PSGL1^lo^Ly6C^lo^ cells are analyzed). Is there a defect not just in Tfh formation but also in their migration to interact with B cells?

7) CD4-cre TGF-bRII^f/f^ mice develop early aggressive autoimmunity and activated T cells appear to emerge over time. One concern is that the Tgf-βR deficient cells are being sort-purified from an activated environment prior to adoptive transfer and this may alter their subsequent responsiveness in vivo.

8) It also needs to be addressed whether the lack of Tgf-β signals during the thymic selection of T cells in CD4-cre TGF-bRII^f/f^ mice might alter their subsequent response to challenge in the periphery.

---

## [Author Response]

*1) Tgf-β is also well established as a factor important for the in vitro generation of Foxp3*^*+*^*CD25*^*+*^
*iTregs and in vivo pTregs from CD4+CD25- cells. In these conditions, Tgf-β is perhaps not directly responsible for inducing CD25 expression, but also does not suppress CD25 expression as iTregs and pTregs both express and are dependent on CD25 expression and IL-2 for their survival. However, in*
Figure 6*, it is clear that CD25 expression is being suppressed by Tgf-β. This discrepancy between the role of Tgf-β in the induction of CD4+CD25+Foxp3*^*+*^
*T-cells and the blocking of CD25 expression seen here needs to be addressed. Is it due to the Stg TCR, which, as the authors demonstrate*, *is not amenable to induction of Foxp3 positive cells?*

We’ve also thought about this issue at length and cannot fully rationalize how TGF-β can sustain (i.e., not suppress) CD25 in one setting, but restrict its expression in another. The most likely explanation is that it depends on the different combinations and doses of cytokines involved in specifying CD4 T cell fates. For example, it is known that T cell differentiation is highly sensitive to TGF-β concentrations and that different types of T cells form according to the concentration of TGF-β (e.g., Tregs vs T_H_17 cells, and Tfh cells) (Zhou, Nature 2008: Schmitt, Nat Immunol, 2014). Additionally, distinct signaling cascades downstream of TGF-βR (e.g., Smad vs. MAPK) may differ based on cell type or TGF-β concentrations (Park, Mol. Immunol., 2007; Giroux, Blood, 2011; Gu, PNAS 2012, among others), which could impact the expression of TGF-β target genes. Furthermore, the specific activity the commercially available recombinant TGF-β is likely to differ considerably between vendors and/or lots, and therefore it is difficult to directly compare one study to another simply based on amount of TGF-β used. Finally, rTGF-β requires activation (cleavage) in a 10mM citric acid solution and the efficiency of activation can be variable from lot-to-lot, not to mention lab-to-lab. For instance, the concentration of TGF-β (10ng/ml) used in our studies was insufficient on its own to maintain FoxP3 protein expression in the absence of exogenous IL-2 in vitro (Figure 7), though it has been demonstrated to do so in other studies (Fantini, Nat Protocols, 2007).Author response image 1.

Further, we feel that IL-2 signaling is an important rheostat between these cell fates since it promotes Treg and suppresses Tfh. It is likely that CD25 expression on iTreg cells is sustained not by TGF-β, but by a dominant positive feed-back loop involving IL-2, STAT5, and Blimp1 (Fontenot et al., Nat Immunol, 2005). Therefore, cells that encounter relatively low amounts of TGF-β in the absence of high amounts of IL-2 will likely not sustain CD25 expression and develop into iTregs (as observed in our hands, see Figure 7). We postulate that this would be an environment more conducive to T_FH_ cell differentiation. It should also be noted that FoxP3 can be induced in activated Stg cells in vitro (as shown in the figure), but it is not expressed in LCMV or influenza-specific Stg cells in vivo. So it is not the TCR signal, per se, that prevents FoxP3 expression in these TCR Tg cells, but likely the combinatorial effects of viral antigen presentation and inflammatory cytokine milieu that is not amenable to induction of FoxP3^+^ Stg Treg cells in vivo. Thus, we do not have a conclusive answer to the reviewer’s question, but we suspect that during LCMV infection, TGF-β may not be produced at a level capable of inducing FoxP3^+^ virus-specific Tregs, but is able to promote T_FH_ cells. We have amended our Discussion to include these important points, highlighted in yellow.

*2) In*
Figure 6*, does this loss of CD25 expression still occur if purified naive T-cells are used alone with polyclonal stimuli such as anti-CD3 and anti-CD28? It is obscure to what extent the effect of the TFGb is direct to the CD4 cells or via other cells in the culture, which includes total splenocytes. This is the most direct demonstration that the effect is due to downregulation of CD25 in particular and not more general changes to activation. Please also include the proliferation data, currently not shown, in*
Figure 6*, that Tgf-β is not affecting activation is a key point*.

This is a great point raised by the reviewers. Because our in vivo analysis of TGF-βR KO CD4 T cells showed heightened CD25, pSTAT5, and pS6 (new addition to Figure 6), we conclude that TGF-β is directly modulating CD25 expression on virus-specific CD4 T cells during infection in vivo. However, in vitro, it is possible that TGF-β was acting on APCs or other cells in the culture to regulate CD25 on the activated CD4 T cells; two situations that are also not mutually exclusive. To this end, we performed the experiment suggested by the reviewers and first purified the CD4 T cells prior to stimulation with anti-CD3+ anti-CD28^+/-^ TGF-β for 2-3 days. In agreement with our in vivo findings, this experiment showed that TGF-β could suppress CD25 and IL-2 responsiveness (i.e., pSTAT5) over time (Figure 8). Thus, we believe that TGF-β is acting directly on the CD4 T cells in vitro to modulate CD25 expression.Author response image 2.

*3) The authors describe the immunological events following viral infection in detail but do not describe any effect on the natural history of viral infection. Are OT-II mice transferred with TGFRII KO or WT CD4 more or less able to resolve infection? Given the recent finding that rapamycin decreases germinal center formation and IgG production but enhances protection from influenza, it is not necessarily clear if the observed effects would be beneficial or detrimental to protection from influenza. The authors suggest that part of the contribution of rapamycin to protection seen in the previous study may have been in enhancing Tfh function but it is hard to see how this matches with the reduction in IgG and GC formation. This will make it possible to make more secure statements regarding the in vivo importance of the work*.

We thank the reviewers for allowing us to address this point. We designed the experiments using OT-II recipients to specifically ask whether T cell-directed TGF-β was required for GC reactions and influenza-specific antibody, but we understand the desire to also assess the pathological outcomes of infection. We felt the experimental approach, while being suitable for comparing the ability of WT and Tgf-βRII KO CD4 T cells to help WT B cells, may present an unphysiological setting for comparing more general aspects of anti-influenza immunity, such as viral control. However, it is still worth examining if a general reduction overall in anti-influenza antibodies may affect the rates of viral clearance. Because our previous experiments went out to day 14 to measure influenza antibodies (and would likely be devoid of virus as influenza is typically cleared within 10 days), we repeated the experiment to examine viral titers at an earlier time point, day 6 p.i. At this relatively early time point, we found that the OT-II mice engrafted with WT T cells had lower amounts (∼10 fold) of influenza viral RNA in their lungs (Figure 9). As this experiment was only done once, we hesitate to make any strong conclusions, and quite honestly, we were surprised to see any effects at all at day 6 because this is relatively early to see the effects of a primary adaptive immune response on control of an influenza A infection (Kamperschroer, J Immunol, 2006; Mozdzanowska, J Immunol, 2000). Nonetheless, this result supports the model that the Tgf-βRII KO cells are less efficient at eliciting anti-viral Ab responses and this correlates with delayed viral control.Author response image 3.

4) The authors interpret their data as evidence that Tgf-β signaling is “required” for Tfh. Can this be justified given that most of their data assesses the relative representation of Tfh versus Th1? Is the major phenomenon being observed that Tgf-β inhibits the Th1 response?

We thank the reviewers for highlighting this important point and allowing us to clarify the differences observed in the roles of TGF-β in Tfh cell formation and function. It is clear based on our data that T cell-directed TGF-β is a critical signal for optimal GC and antibody responses during influenza infection. And although the effect of TGF-β deletion on T_FH_ cell formation and function is profound, as noted by the reviewers in point 6 below, the functional defects may be even stronger than the phenotypic defects. Thus, we conclude that TGF-β is a necessary component of Tfh cell function during influenza infection. Many of the attributes we use to detect Tfh cell formation (CXCR5, PD-1, Ly6C, PSGL1) were affected by deletion of Tgf-βRII, but they were not fully abrogated. Therefore, we agree that we should be more conservative in our description of the role of TGF-β in T_FH_ cell formation. In support of this, TGF-β preferentially induced expression of CXCR5 and ICOS more than Bcl6 and IL-21 in human CD4 T cells (Schmitt et al., Nat Immunol, 2014), and suggests that TGF-β may contribute to part, but not all, of the Tfh cell program. We have revised the text accordingly. To better understand how TGF-β contributes to T_FH_ cell development and function, we are preparing to compare the global gene expression profiles of the flu-specific WT and KO CD4 T cells via RNA-seq. This could help us identify which T_FH_ signature genes are most affected by TGF-β signaling.

*5) How do the authors explain the observation that “Tfh precursor cells” are impaired at d3 but normal by d8 (*Figure 2—figure supplement 1*)? Why do other signals compensate for Tgf-β in the setting of LCMV infection but not influenza?*

We find this discrepancy particularly fascinating and would like to point out that this is not the first example of a signal that is required for a precursor population but not full effector T cell differentiation. Shane Crotty’s lab has found that IL-6 is an important signal for Tfh precursors (Choi et al., J. Immunol., 2013), but is dispensable for full Tfh effector cells during LCMV infection (Poholek, J. Immunol, 2010). We hypothesize that a determining factor between LCMV and influenza infections lies within the context of the viral infection (mucosal vs. systemic), and that TGF-β is more abundant in the lung and medLN and plays a more prominent role in mucosal immune responses, but we currently only have circumstantial evidence for this. Our future studies will examine this hypothesis more closely by comparing additional systemic and mucosal infections, as well as different mucosal tissues such as the gut.

*6) The B cell analysis (*Figure 4*) is very clear, perhaps more striking than the Tfh data (particularly where PSGL1*^*lo*^*Ly6C*^*lo*^
*cells are analyzed). Is there a defect not just in Tfh formation but also in their migration to interact with B cells?*

This is a very important point, and we thank the reviewers for ensuring that we clarify this critical piece of data in our manuscript. As one can see in Figure 2, we indeed find fewer TGF-βRII KO STg cells found within PNA^+^ GC. Because both the downregulation of PSGL1 (as well as CCR7) and the upregulation of CXCR5 are important for their migration and retention in the GC, we believe that the aberrant expression of these molecules may preclude them from getting into or staying in the GC to provide B cell help. Whether they traffic there but do not remain or are impaired in their homing is also an interesting question, but one that would be best addressed using intravital imaging which is an expertise not present in our lab.

*7) CD4-cre TGF-bRII*^*f/f*^
*mice develop early aggressive autoimmunity and activated T cells appear to emerge over time. One concern is that the Tgf-βR deficient cells are being sort-purified from an activated environment prior to adoptive transfer and this may alter their subsequent responsiveness in vivo*.

We agree with the reviewers on this point and we worked very hard to design our experiments appropriately in order to address this concern. When we first identified the TGF-β gene signature in WT Tfh cells, we sought advice from Richard Flavell at Yale who created these mice and has worked with this strain extensively to seek technical guidance on this exact point. They informed us to always sort on CD44^lo^ naïve phenotype T cells, which we have done in all experiments. Furthermore, we crossed the TGF-βRII^f/f^ mice to the Smarta TCR transgenic (Stg) mice, which will slow the rate of any autoimmunity by restricting the repertoire (Sanjabi et al., J. Immunol. Methods, 2010). Additionally, for the experiments involving the adoptive transfer of polyclonal CD4 T cells (Figures 3 and 4) we used the TGF-βRII^f/f^ distal Lck-cre mice because it has been reported that while nearly all the conventional T cells (i.e., non-T_reg_ cells) lack TGF-βRII in this model, a fraction of CD4 T cells do escape cre-mediated deletion and develop into Tregs with the TGF-βRII intact. As such, the TGF-βRII^f/f^ distal Lck-cre mice have no signs of autoimmunity during the first 18 weeks of life (Zhang, Nat Immunol, 2012).

*8) It also needs to be addressed whether the lack of Tgf-β signals during the thymic selection of T cells in CD4-cre TGF-bRII*^*f/f*^
*mice might alter their subsequent response to challenge in the periphery*.

TGF-β signaling in the thymus can indeed impact T cell selection, as demonstrated clearly by Ouyang et al. (Immunity, 2013). It appears that TGF-β signals regulate at least 2 facets of T cell homeostasis. First, TGF-β promotes the selection of low affinity T cells in the thymus, and second, TGF-β regulates peripheral survival of naïve CD4 T cells as well as Treg and Th17 cells (Ouyang, Immunity, 2013; Ouyang, Immunity, 2010; Li, Immunity, 2006). Although we did not go to the same lengths to assess these attributes for TGF-βR^f/f^ CD4-cre Stg cells, we did not observe a survival defect for TGF-βRII KO Stg cells in the spleen or lymph nodes of young adult mice. Therefore, it appears that the overexpression of the high affinity Stg TCR bypasses any need for TGF-β signals for naïve T cell selection and survival and we believe that our comparisons of WT and TGF-βR KO naïve CD4 T cells are fair.